# Fast Mixing of Stochastic Gradient Descent with Normalization and Weight Decay

**Zhiyuan Li**
Princeton University
zhiyuanli@cs.princeton.edu

**Tianhao Wang**
Yale University
tianhao.wang@yale.edu

**Dingli Yu**
Princeton University
dingliy@cs.princeton.edu

## Abstract

We prove the Fast Equilibrium Conjecture proposed by Li et al. [1], *i.e.*, stochastic gradient descent (SGD) on a scale-invariant loss (*e.g.*, using networks with various normalization schemes) with learning rate $\eta$ and weight decay factor $\lambda$ mixes in function space in $\widetilde{\mathcal{O}}(1/(\eta\lambda))$ steps, under two standard assumptions: (1) the noise covariance matrix is non-degenerate and (2) the minimizers of the loss form a connected, compact and analytic manifold. The analysis uses the framework of Li et al. [2] and shows that for every $T > 0$, the iterates of SGD with learning rate $\eta$ and weight decay factor $\lambda$ on the scale-invariant loss converge in distribution in $\ln(1 + T\lambda/\eta)/(4\eta\lambda)$ iterations as $\eta\lambda \to 0$ while satisfying $\eta \leq O(\lambda) \leq O(1)$. Moreover, the evolution of the limiting distribution can be described by a stochastic differential equation that mixes to the same equilibrium distribution for every initialization around the manifold of minimizers as $T \to \infty$.

## 1 Introduction

Generalization in modern deep learning has significantly deviated from classical learning theory due to the vast overparametrization in deep neural networks and is underlain by the implicit bias of training algorithms [3]. Instead of decreasing the training objective as fast as possible, the training algorithm and its hyperparameters are often tuned for good implicit bias, *i.e.*, the ability to pick empirical minimizers with good generalization among various different minimizers. Sometimes good implicit bias occurs at the cost of less efficient optimization, including the usage of large learning rates (LR) [4] or small batch size [5, 6]. Thus the training objective alone is not an effective measure of the entire training progress. In other words, behind the minimization of the training objective, there potentially exists some *hidden progress*, and the evolution of the model therein plays a crucial role in the implicit bias.

The current paper aims to provide a better theoretical understanding of such hidden progress for neural networks equipped with normalization layers (e.g., BatchNorm [7], LayerNorm [8], and others [9–13]) trained by Stochastic Gradient Descent (SGD) with Weight Decay (WD), dubbed SGD+WD. For learning rate (LR) $\eta$ and WD factor $\lambda$, we formulate SGD+WD as

$$x_{\eta,\lambda}(k+1) = (1 - \eta\lambda)x_{\eta,\lambda}(k) - \eta\nabla L_{\xi_k}(x_{\eta,\lambda}(k)) \tag{1}$$

where $x_{\eta,\lambda}(k) \in \mathbb{R}^D$ is the parameter after $k$ iterations, and $L_{\xi_k}$ is the loss over the $\xi_k$-th sample with each $\xi_k$ being sampled independently and uniformly randomly across all training data. In particular, we are interested in explaining the following phenomenon:

> *Longer training with SGD+WD after LR decay improves final test accuracy of normalized networks.*

We demonstrate such phenomenon in Figure 1, where test accuracy after LR decay keeps improving when training accuracy plateaus. In an extreme case, Li et al. [1] empirically showed that the test

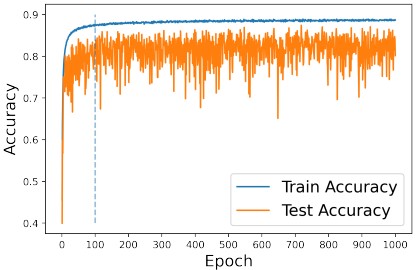

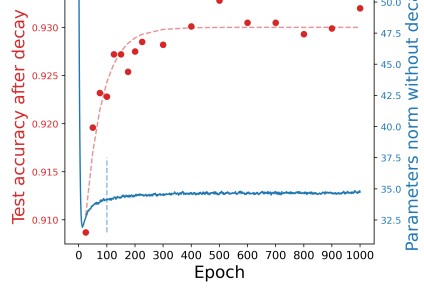

(a) Train and test accuracy for CIFAR-10 training with $\eta = 0.8, \lambda = 5 \cdot 10^{-4}$.

(b) Test accuracy after LR decay and the total norm of parameters before LR decay.

Figure 1: The train and test accuracy plateaus after parameter norm convergence within 100 epochs, but the generalization of SGD iterate after LR decay keeps improving. Figure 1a shows the train and test accuracy of scale invariant PreResNet trained by SGD+WD on CIFAR-10 with standard data augmentation. Each red dot in Figure 1b represents the test accuracy of model which decays LR to $10^{-3}$ at the corresponding epoch. The test accuracy is evaluated until achieving full training accuracy after LR decay.

accuracy of ResNet can still improve after maintaining nearly full training accuracy for thousands of epochs when trained by SGD+WD on CIFAR10. Such phenomenon is also demonstrated for a standard decoder-only Transformer trained by Adam on small arithmetic datasets and is named 'grokking' by Power et al. [14], where the validation accuracy can increase from random guess to full accuracy long after the almost perfect fitting of the training data.

Based on theoretical derivations, Li et al. [1] further proposed the *Fast Equilibrium Conjecture* (Conjecture 1.1), which informally says that for the normalized model trained by SGD+WD, such hidden progress happens in $\widetilde{O}(1/(\eta\lambda))$ steps and the model converges to an equilibrium, and since then, further training can no longer improve the final test accuracy. A recent line of works [15, 16, 2] show that gradient noise in stochastic gradient can cause a higher order regularizing effect and improve generalization even when training loss is close to 0. In particular, Li et al. [2] proposed a mathematical framework for characterizing the implicit bias of SGD in the time scale of $O(1/\eta^2)$. Under such a time scale, the hidden progress of SGD is shown to be described by a Stochastic Differential Equation (SDE) termed the limiting diffusion, which can then be used to rigorously prove its generalization benefit in some cases (see Section 6 in [2]).

However, the $O(1/\eta^2)$ rate given by Li et al. [2] is not applicable to networks with normalization layers and WD, because the assumptions on the loss landscape made in [2] that minimizers of training loss connect as a manifold fail to hold for networks with normalization layers and WD, or more broadly, for all *scale invariant* loss (see Definition 2.2) with $\ell_2$ regularization. Here a loss $L$ is scale invariant means that $L(Cx) = L(x)$ for any $C > 0$ and parameter $x \neq 0$, which is a consequence of normalization layers. The assumption of manifold of minimizers fails because any $\ell_2$-regularized scale invariant loss has no local minimizer, not to mention the manifold of minimizers. To see this, simply note that for every $x$ where scale invariant loss is well-defined, *i.e.*, $x \neq 0$, reducing its norm while keeping the direction of $x$ strictly decreases $\ell_2$ regularized scale invariant loss. Moreover, the loss landscape becomes unboundedly sharp around the origin. These drastic changes to loss landscape induced by normalization layers could lead to bizarre training dynamics beyond the scope of standard optimization viewpoint, *e.g.*, deep neural networks with normalization can even be trained with an exponentially increasing LR schedule [17].

## 1.1 Our Results

In this paper, we show that for networks with normalization trained by SGD+WD the $\widetilde{O}(1/(\eta\lambda))$ rate is indeed the correct time scale for the aforementioned hidden progress and deliver a partial proof to the Fast Equilibrium Conjecture proposed by Li et al. [1]. The key observation here is that we need to rescale the SGD+WD dynamics both in time and parameter norm by leveraging the scale invariance of loss, so that the framework in [2] can again be applied to achieve an SDE-based characterization for the hidden progress. Our rescalings are motivated by the analysis in [1] for the parameter norm convergence which happens in $\widetilde{O}(1/(\lambda\eta))$ steps.

Before stating the main theorem, we will first introduce some notations and restate the Fast Equilibrium Conjecture. Let $F_z(x)$ be the output of a scale invariant neural network with parameter $x$ on data $z$, *i.e.*, $F_z(x) = F_z(Cx)$ for any parameter $x$, input data $z$ and constant $C > 0$. In other words, the output of the network only depends on the direction of parameter, shorthanded as $\overline{x} := x/\|x\|_2$.

Let $\Xi$ be the total number of training data and $L(x) = \frac{1}{\Xi}\sum_{\xi=1}^{\Xi} L_\xi(x)$ be the empirical training loss. Denote $\sigma(x) = (\sigma_1(x), \ldots, \sigma_\Xi(x))$ where each $\sigma_\xi(x) := (\nabla L_\xi(x) - \nabla L(x))/\sqrt{\Xi}$. Then we can rewrite SGD+WD (Equation (1)) as

$$x_{\eta,\lambda}(k+1) = (1-\eta\lambda)x_{\eta,\lambda}(k) - \eta\big(\nabla L(x_{\eta,\lambda}(k)) + \sqrt{\Xi}\cdot\sigma_{\xi_k}(x_{\eta,\lambda}(k))\big). \tag{2}$$

Let $\{W(t)\}_{t\geq 0}$ be a $\Xi$-dimensional Brownian motion. As a common approach to analyzing SGD, the canonical SDE approximation of SGD+WD (Equation (2)) is

$$dX_{\eta,\lambda}(t) = -\eta\nabla L(X_{\eta,\lambda}(t))dt - \eta\lambda X_{\eta,\lambda}(t)dt + \eta\sigma(X_{\eta,\lambda}(t))dW(t) \tag{3}$$

The Fast Equilibrium Conjecture is stated below. The convergence rate is much faster than the $e^{-\Theta(1/\eta)}$ global mixing time of Langevin dynamics [18] and thus the conjecture gets its name.

**Conjecture 1.1** (Fast Equilibrium Conjecture, Li et al. [1])**.** *Suppose $X_{\eta,\lambda}(t)$ is a solution of (3), then for any input $z$, $F_z(X_{\eta,\lambda}(t))$ converges to the same equilibrium distribution independent of the initial parameter $x_{\text{init}}$ in $\widetilde{O}(1/(\eta\lambda))$ time.*

We note that the above conjecture is implied by the convergence of the distribution of the parameter direction $\overline{X}_{\eta,\lambda}(t)$. Next, the main theorem of this paper is stated informally below.

**Theorem 1.2** (Informal version of Theorem 5.5)**.** *Suppose $\Gamma$ is a connected manifold consisting only of local minimizers of $L$. Under some regularity assumptions, there is an open neighborhood $U$ of $\Gamma$, such that for any initialization $x_{\text{init}} \in U$ and $T > 0$, as $\eta\lambda \to 0$ with $\eta \leq O(\lambda) \leq O(1)$, both $\bar{x}_{\eta,\lambda}\Big(\Big\lfloor\frac{\ln(\frac{2\lambda}{\eta}(e^{2T}-1)+1)}{4\eta\lambda}\Big\rfloor\Big)$ and $\overline{X}_{\eta,\lambda}\Big(\frac{\ln(\frac{2\lambda}{\eta}(e^{2T}-1)+1)}{4\eta\lambda}\Big)$ converge in distribution to the same distribution denoted by $\mu_{T,x_{\text{init}}}$. Moreover, as $T \to \infty$, $\mu_{T,x_{\text{init}}}$ weakly converges to the same equilibrium distribution for every $x_{\text{init}} \in U$.*

The **main contributions** of this paper are summarized as follows:

1. We give a SDE-based characterization (Theorem 4.4) for the limiting dynamics of SGD+WD for a scale invariant stochastic loss in the limit of $\eta\lambda \to 0$ with $\eta = O(\lambda)$ and $\lambda = O(1)$. By introducing a novel time-rescaling tailored to the scale invariant loss and weight decay, our analysis adapts the framework proposed by Li et al. [2].

2. We show that SGD without WD for a stochastic scale invariant loss has the same limiting dynamics as that of SGD+WD, but is exponentially slower (see Theorem 3.2). This is consistent to the empirical observation that turning on WD for SGD for scale invariant loss helps generalization [19, 1].

3. Under the assumption of all minimizers forming a manifold and noise being non-degenerate in the tangent space of the manifold, we show that from any initialization, the limiting dynamics of SGD+WD converges to a unique stationary distribution (see Theorem 5.4 and Theorem 5.5). This delivers a partial proof to the Fast Equilibrium Conjecture in [1].

4. Though our convergence result is asymptotic, we verify in simplified settings that the phenomena predicted by our theory happens with LR $\eta$ and WD factor $\lambda$ of practical scale (see Section 6 for details of experiments). We also show empirically that the mixing process exists in practical settings, and is beneficial for generalization.

## 2 Preliminary

**Notations.** We denote by $\mathbb{N}$ the set of all nonnegative integers and $\mathbb{R}_+$ the set of all nonnegative real numbers. For any $k \in \mathbb{N}$, we denote by $\mathcal{C}^k$ the set of all $k$ times continuously differentiable functions. For any vector $u \in \mathbb{R}^D$, we denote its $i$-th coordinate by $u_i$. For any mapping $F: \mathbb{R}^D \to \mathbb{R}^D$, we denote the *Jacobian* of $F$ at $x$ by $\partial F(x) \in \mathbb{R}^{D\times D}$ where the $(i,j)$-th entry is $\partial_j F_i(x)$. We also use $\partial F(x)[u]$ and $\partial^2 F(x)[u,v]$ to denote the first and second order directional derivatives of $F$ at $x$ along the derivation of $u$ (and $v$). With a slight abuse of notation, we view $\partial^2 F$ as a linear mapping on $\mathbb{R}^{D\times D}$ such that $\partial^2 F(x)[A] = \sum_{i,j=1}^{D} \partial^2 F(x)[e_i, e_j]A_{ij}$, for any $A \in \mathbb{R}^{D\times D}$. For any submanifold $\Gamma \subset \mathbb{R}^D$ and $x \in \Gamma$, we denote by $T_x(\Gamma)$ the tangent space of $\Gamma$ at $x$. We denote by $\mathbb{1}_\xi \in \mathbb{R}^\Xi$ the one-hot vector where the $\xi$-th coordinate is 1, and $\mathbb{1}$ denotes the all 1 vector. We say $K \subset \mathbb{R}^D$ is a *cone* if and only if $0 \notin K$ and $\forall \alpha > 0, \alpha K \subseteq K$.

Recall that the training loss $L : \mathbb{R}^D \to \mathbb{R}$ is defined as $L = \frac{1}{\Xi} \sum_{i=1}^{\Xi} L_i$ where $L_i$ is the loss on the $i$-th sample. With vast overparametrization in modern machine learning models, multiple minimizers can exist and form a manifold [20, 21]. Thus following Fehrman et al. [22], Li et al. [2], Arora et al. [23], we make the assumption below throughout this paper.

**Assumption 2.1.** Each loss function $L_i : \mathbb{R}^D \to \mathbb{R}$ is a $\mathcal{C}^4$ function. $\Gamma$ is a $(D - M)$-dimensional $\mathcal{C}^2$-submanifold of $\mathbb{R}^D$ for some integer $M \in [0, D - 1]$, where each $x \in \Gamma$ is a local minimizer of $L$ and $\text{rank}(\nabla^2 L(x)) = M$ for all $x \in \Gamma$.

Note that $\nabla^2 L(x)$ must have zero eigenvalues in the tangent space of $\Gamma$ at $x$, we are indeed assuming the Hessian $\nabla^2 L$ attains the maximal rank everywhere on the manifold $\Gamma$.

In this paper, we are interested in the behavior of SGD+WD with each $L_i$ being scale invariant, or equivalently, 0-homogenous. We note that the level sets of scale invariant functions are always cones, which will be used frequently in our analysis. To this end, we make Assumption 2.3.

**Definition 2.2** (Homogeneous Functions). We say a function $f : \mathbb{R}^D \setminus \{0\} \to \mathbb{R}^m$ is a $k$-*homogeneous* for some $k \in \mathbb{R}$ if and only if for all $x \in \mathbb{R}^D \setminus \{0\}$ and $\alpha > 0$, $f(\alpha x) = \alpha^k f(x)$. Specifically, we say a function $f$ is *scale invariant* if and only if it is 0-homogeneous.

**Assumption 2.3.** $L_i$ is scale invariant for each $1 \le i \le \Xi$ and $\Gamma$ is a cone.

Below are two useful properties of homogeneous functions, whose proofs follow from directly applying chain rules.

**Lemma 2.4.** *For any $l \in \mathbb{N}$ and $k$-homogeneous function $f$, $\nabla^l f$ is $(k - l)$-homogeneous.*

**Lemma 2.5** (Euler's Theorem for Homogeneous Functions). *For any real-valued $k$-homogeneous function $f$, $\langle x, \nabla f(x) \rangle = k f(x)$. Specifically, if $f$ is scale invariant, $\langle x, \nabla f(x) \rangle \equiv 0$.*

Recall that $\sigma_i(x) = \frac{1}{\sqrt{\Xi}}(\nabla L_i(x) - \nabla L(x))$, so the noise function $\sigma$ is $(-1)$-homogeneous and thus the noise covariance $\Sigma(x) = \sigma(x)\sigma(x)^\top$ is $(-2)$-homogeneous.

Next, the following notion of limiting map of gradient flow plays a key role in our analysis.

**Definition 2.6.** For any $x \neq 0$, we define the *gradient flow* governed by $-\nabla L$ as the unique solution of $\phi(x, t) := x - \int_0^t \nabla L(\phi(x, s)) \mathrm{d}s$ for $t \ge 0$ and denote its associated limiting map by $\Phi(x) = \lim_{t \to \infty} \phi(x, t)$ whenever the limit exists.

Throughout the paper, we use $U$ to denote the attraction set of $\Gamma$ under gradient flow, that is, $U = \{x \in \mathbb{R}^D \mid \Phi(x) \text{ is well-defined and } \Phi(x) \in \Gamma\}$. By Lemma B.15 of Arora et al. [23], $U$ is open and $\Phi$ is $\mathcal{C}^2$ in $U$ with $\nabla^2 \Phi$ being locally lipschitz.

**Lemma 2.7.** *Under Assumption 2.3, $U$ is a cone and $\Phi$ is 1-homogeneous in $U$.*

Li et al. [2] established several important properties of $\Phi$ by relating the derivatives of $\Phi$ to those of $L$, and in particular, we recall the following characterization of $\partial \Phi$.

**Lemma 2.8** (Lemma 4.3, [2]). *For any $x \in \Gamma$, $\partial \Phi(x) \in \mathbb{R}^{D \times D}$ is the orthogonal projection matrix onto the tangent space $T_x(\Gamma)$. As a consequence, for any $x \in \Gamma$, Assumption 2.3 $\implies x \in T_x(\Gamma) \implies \partial \Phi(x) x = x$.*

## 2.1 Limiting Diffusion on The Manifold of Local Minimizers

We recap the notion of *Katzenberger processes* proposed by Li et al. [2] and the characterization of the corresponding limiting diffusion based on Katzenberger's theorems [24]. In this subsection we only assume Assumption 2.1, but not Assumption 2.3.

**Definition 2.9** (Uniform metric). The *uniform metric* between two functions $f, g : [0, \infty) \to \mathbb{R}^D$ is defined to be $d_U(f, g) = \sum_{T=1}^{\infty} 2^{-T} \min\{1, \sup_{t \in [0, T)} \|f(t) - g(t)\|_2\}$.

For each $n \in \mathbb{N}$, let $A_n : \mathbb{R}_+ \to \mathbb{R}_+$ and $B_n : \mathbb{R}_+ \to \mathbb{R}_+$ be two non-decreasing functions with $A_n(0) = B_n(0) = 0$, and $\{Z_n(t)\}_{t \ge 0}$ be a $\mathbb{R}^\Xi$-valued stochastic process. In our context of SGD+WD, given loss function $L : \mathbb{R}^D \to \mathbb{R}$, noise function $\sigma : \mathbb{R}^D \to \mathbb{R}^{D \times \Xi}$ and initialization $x_{\text{init}} \in U$, we call the following stochastic process (4) a *Katzenberger process*

$$X_n(t) = x_{\text{init}} + \int_0^t \sigma(X_n(s))\mathrm{d}Z_n(s) + \int_0^t X_n(t)\mathrm{d}B_n(s) - \int_0^t \nabla L(X_n(s))\mathrm{d}A_n(s) \quad (4)$$

if as $n \to \infty$ the following conditions are satisfied:

1. $A_n$ increases infinitely fast, i.e., $\forall \epsilon > 0, \inf_{t \geq 0}(A_n(t + \epsilon) - A_n(t)) \to \infty$;

2. $B_n(t)$ converges to $c \cdot t$ in uniform metric for some constant $c$.

3. $Z_n$ converges in distribution to $(I_\Xi - \frac{1}{\Xi}\mathbb{1}\mathbb{1}^\top)W$ in uniform metric where $W$ is a $\Xi$-dimension standard Brownian motion;

**Theorem 2.10** (Adapted from Theorem 4.6 in Li et al. [2]). *Given a Katzenberger process $\{X_n(\cdot)\}_{n \in \mathbb{N}}$, if SDE (5) has a global solution $Y$ in $U$ with $Y(0) = \Phi(x_{\text{init}})$, then for any $t > 0$, $X_n(t)$ converges in distribution to $Y(t)$ as $n \to \infty$.*

$$Y(t) = \Phi(x_{\text{init}}) + \int_0^t c\partial\Phi(Y(s))Y(s)\mathrm{d}s + \int_0^t \partial\Phi(Y(s))\sigma(Y(s))\mathrm{d}W(s)$$

$$+ \int_0^t \frac{1}{2}\partial^2\Phi(Y(s))[\Sigma(Y(s))]\mathrm{d}s. \tag{5}$$

We note that the global solution always exists if the manifold $\Gamma$ is compact. Our notion of Katzenberger process and theorem statement is slightly more general than those in Li et al. [2] to handle the weight decay. However, our formulation is still under the original framework of Katzenberger [24] and the proof in Li et al. [2] can be easily adapted to Theorem 2.10.

# 3 Warm-up: Simultaneous Limit Case

As a warm-up, we first consider the setting where $\eta, \lambda \to 0$ simultaneously with $\frac{\lambda}{\eta} \equiv C$ for some constant $C \geq 0$. In this special regime, we do not need to use the scale invariance property of the loss, and we can directly apply Theorem 2.10 to obtain the limiting diffusion of SGD+WD. Nonetheless, we will see the benefit of weight decay as a source of acceleration. While for the general case of $\eta\lambda \to 0$ that will be considered in Section 4, we need to carefully design a time rescaling by calibrating with the dynamics of parameter magnitude, so that under the new scaling the dynamics can still be understood as a Katzenberger process.

Now, recall the SGD+WD updates in Equation (2), and let us fix $x_{\eta,\lambda}(0) = x_{\text{init}}$ for some $x_{\text{init}} \in U$. Define $\check{X}_{\eta,\lambda}(t) = x_{\eta,\lambda}(\lfloor t/\eta^2 \rfloor)$, which is roughly equivalent to SDE (3) with $1/\eta^2$ times acceleration, and we can rewrite the discrete-time update of $x_{\eta,\lambda}$ as

$$\check{X}_{\eta,\lambda}(t) = x_{\text{init}} + \int_0^t \sigma(\check{X}_{\eta,\lambda}(s))\mathrm{d}Z_{\eta,\lambda}(s) + \int_0^t \check{X}_{\eta,\lambda}(s)\mathrm{d}B_{\eta,\lambda}(s) - \int_0^t \nabla L(\check{X}_{\eta,\lambda}(s))\mathrm{d}A_{\eta,\lambda}(s) \tag{6}$$

where $A_{\eta,\lambda}, B_{\eta,\lambda}$ and $Z_{\eta,\lambda}$ are defined by

$$A_{\eta,\lambda}(t) = \eta\lfloor t/\eta^2 \rfloor, \quad B_{\eta,\lambda}(t) = \lambda\eta\lfloor t/\eta^2 \rfloor, \quad Z_{\eta,\lambda}(t) = \eta\sum_{k=1}^{\lfloor t/\eta^2 \rfloor}\sqrt{\Xi}\left(\mathbb{1}_{\xi_k} - \frac{1}{\Xi}\mathbb{1}\right). \tag{7}$$

Note that $A_{\eta,\lambda}(t)$ is roughly $t/\eta$ which becomes very large for small $\eta$, thus the negative gradient part will drive $\check{X}_{\eta,\lambda}(t)$ rapidly towards the manifold $\Gamma$ and force $\check{X}_{\eta,\lambda}(t)$ to stay close to $\Gamma$ after that. On the other hand, as $\eta \to 0$, $B_{\eta,\lambda}(t)$ will converge to $Ct$ and $Z_{\eta,\lambda}$ will weakly converge to a Brownian motion, and these terms make up the slow dynamics of SGD. More precisely, we have the following lemma summarizing the properties of these integrators, which shows Equation (6) is a valid Katzenberger process.

**Lemma 3.1.** *Let $A_{\eta,\lambda}, B_{\eta,\lambda}$ and $Z_{\eta,\lambda}$ be as defined in Equation (7). Then as $\eta, \lambda \to 0$ with $\frac{\lambda}{\eta} \equiv C$, it holds that (1) $A_{\eta,\lambda}$ increases infinitely fast, (2) $B_{\eta,\lambda}(t)$ converges to $Ct$ in uniform metric and (3) $Z_{\eta,\lambda}$ converges in distribution to $(I_\Xi - \frac{1}{\Xi}\mathbb{1}\mathbb{1}^\top)^{1/2}W$ in uniform metric where $\{W(t)\}_{t \geq 0}$ is the $\Xi$-dimensional standard Brownian motion.*

Therefore, a direct application of Theorem 2.10 yields the limiting diffusion in this case.

**Theorem 3.2.** *Under Assumption 2.1, let $x_{\eta,\lambda}(0) \equiv x_{\text{init}} \in U, \forall \eta, \lambda > 0$ in SGD+WD (2). Consider*

$$\mathrm{d}Y_C(t) = -C\partial\Phi(Y_C)Y_C\mathrm{d}t + \frac{1}{2}\partial^2\Phi(Y_C)[\Sigma(Y_C)]\mathrm{d}t + \partial\Phi(Y_C)\sigma(Y_C)\mathrm{d}W(t) \tag{8}$$

*where $\{W(t)\}_{t \geq 0}$ is the standard Brownian motion in $\mathbb{R}^\Xi$. Suppose SDE (8) has a global solution $Y_C$ in $U$ for some $C \geq 0$ with $Y_C(0) = \Phi(x_{\text{init}})$, then $x_{\eta,\lambda}(\lfloor t/\eta^2 \rfloor)$ converges in distribution to*

$Y_C(t)$ as $\lambda, \eta \to 0$ with $\frac{\lambda}{\eta} \equiv C$. Also, under Assumption 2.3, SDE (8) with any $C' \geq 0$ has a global solution $Y_{C'}$ in $U$ with $Y_{C'}(0) = \Phi(x_{\text{init}})$. Moreover, $Y_{C'}(t) \overset{d}{=} Y_0(\frac{e^{4C't}-1}{4C'})e^{-C't}$.

**Remark 3.3.** *Theorem 3.2 shows that when there is no WD, the limiting diffusion is still the same as that with WD, but exponentially slower than that with WD in the regime of LR $\eta$ and WD factor $\lambda$ going to zero with a fixed ratio.*

## 4 Limiting Diffusion for The General Case

In the previous section, we showed that the limiting diffusion exists when $\eta$ and $\lambda$ go to zero with a fixed ratio. However, the situation is more complicated in the general case where we drop the assumption of $\eta/\lambda$ being fixed. Below we first explain the challenges in analysis and our solution for this general regime. Then we present the continuous-time analysis for SDE and the discrete-time analysis for SGD+WD. Our analysis applies for all the cases when $\eta\lambda \to 0$ with $\eta = O(\lambda) = O(1)$.

**Challenges for the General Case.** A concrete example for the challenge is when $\eta \to 0$ and $\lambda$ be fixed as a constant, which is also the most natural and practical setting. We quickly find ourselves in a dilemma if we still want to apply Katzenberger's theorem [24], or its simplified version Theorem 2.10: If we view WD or $\ell_2$ regularization as the 'fast' past of the dynamics, that is, a part of the loss function, then there is no minimizer for the $\ell_2$ regularized scale-invariant loss and thus it doesn't satisfy the condition of Katzenberger's theorem; if we view WD as some 'slow' dynamics and formulate it as $\frac{\lambda}{\eta} X_{\eta,\lambda} dB_{\eta,\lambda}(t)$ in Equation (6), then unlike the simultaneous limit case, $\frac{\lambda}{\eta}$ doesn't necessarily have a limit, and thus the condition of Katzenberger's theorem is again not met.

The above dilemma reflects two different roles of WD in early and late phase of training: in the early phase, when the norm is large, WD is more like a part of the loss function that executes the $\ell_2$ regularization. In contrast, in the late phase of SGD training, especially when the norm square of parameters has stabilized to some value, *i.e.*, $\|x_{\eta,\lambda}(t)\|_2^2 \propto \sqrt{\frac{\eta}{\lambda}}$ (*e.g.* Figure 1b), WD should be viewed as the 'slow' dynamics and we can apply the analysis in the simultaneous limit case. This is because by the scale-invariance of loss $L$, Equation (2) can be rewritten as the following form, Equation (9), with $\tilde{\eta} = \sqrt{\eta\lambda}, \tilde{\lambda} = \sqrt{\eta\lambda}, \tilde{x}_{\tilde{\eta},\tilde{\lambda}} = (\frac{\lambda}{\eta})^{1/4} x_{\eta,\lambda}$.

$$\tilde{x}_{\tilde{\eta},\tilde{\lambda}}(k+1) = (1 - \tilde{\eta}\tilde{\lambda})\tilde{x}_{\tilde{\eta},\tilde{\lambda}}(k) - \tilde{\eta}(\nabla L(x_{\tilde{\eta},\tilde{\lambda}}(k)) + \sqrt{\Xi}\sigma_{\tilde{\xi}_k}(x_{\tilde{\eta},\tilde{\lambda}}(k))) \tag{9}$$

With such a rescaling, we successfully make the norm of parameters in constant scale, that is, $\|\tilde{x}_{\tilde{\eta},\tilde{\lambda}}\|_2^2 = \sqrt{\frac{\lambda}{\eta}}\|x_{\eta,\lambda}\|_2^2 = \Theta(1)$ and thus we can apply Katzenberger's theorem. Note that we cannot do this in the early phase of SGD+WD as we start from a fixed initialization and such a rescaling will change the magnitude of the initilaization.

**Our Strategy for Analysis.** To overcome the above dilemma, our core strategy is to introduce a novel combination of parameter rescaling $R_{\eta,\lambda}$ (Equation (11)) and time rescaling $\tau_{\eta,\lambda}$ (Equation (12)) which smoothly interpolates the early and late regime. Because the rescalings are adaptive to the norm of the parameter along the training trajectory, they allow us to apply Katzenberger's theorem on the rescaled dynamics $\frac{X_{\eta,\lambda}(\tau_{\eta,\lambda}^{-1}(t))}{R_{\eta,\lambda}(\tau_{\eta,\lambda}^{-1}(t))}$. (See formal statements in Theorem 4.1). Compared with the ordinary SGD without WD studied in Li et al. [2] where the time rescaling is set to be a fixed acceleration by $1/\eta^2$ times, here the design of the time rescaling is more complicated.

Since the norm has no effect on the loss value but only affects the speed, we need to consider the dynamics of the parameter direction. To do so, we need to normalize the iterates properly. However, when the trace of the noise covariance is not constant, in general it is hard to find a close-form solution for $\|X_{\eta,\lambda}(t)\|_2$, but we can approximate it using the former special case. In specific, recall the canonical SDE approximation in (3). Li et al. [1] proved that the dynamics of $\|X_{\eta,\lambda}(t)\|_2^2$ is

$$d\|X_{\eta,\lambda}(t)\|_2^2 = -2\eta\lambda\|X_{\eta,\lambda}(t)\|_2^2 + \eta^2 \operatorname{tr}(\Sigma(X_{\eta,\lambda}(t))dt.$$

Suppose $\operatorname{tr}(\Sigma(x)) \equiv \hat{\sigma}^2/\|x\|_2^2$ for some $\sigma > 0$, then the above further simplifies into $d\|X_{\eta,\lambda}(t)\|_2^2 = -2\eta\lambda\|X_{\eta,\lambda}(t)\|_2^2 dt + \frac{\eta^2\hat{\sigma}^2}{\|X_{\eta,\lambda}(t)\|_2^2} dt$, which admits a closed-form solution:

$$\|X_{\eta,\lambda}(t)\|_2^4 = \frac{\eta\hat{\sigma}^2}{2\lambda} + e^{-4\lambda\eta t}\left(\|X_{\eta,\lambda}(0)\|_2^4 - \frac{\eta\hat{\sigma}^2}{2\lambda}\right). \tag{10}$$

This implies that the norm of the weights at the equilibrium is of order $(\eta/\lambda)^{1/4}$. Moreover, Equation (10) reflects the scaling of the norm of the iterates in terms of $\eta$ and $\lambda$ as we will see later.

## 4.1 Continuous-time Analysis for SDE

We first consider the continuous-time case of the SDE approximation (3). The main result is summarized in Theorem 4.1, which shows that the limiting diffusion exists for SDE with a suitable non-linear rescaling.

As mentioned in the previous discussion, we consider a scaling function $R_{\eta,\lambda}(t)$ inspired by the norm dynamics for the special case in Equation (10):

$$R_{\eta,\lambda}(t) = \left( \frac{\eta}{2\lambda} + e^{-4\eta\lambda t} \left( 1 - \frac{\eta}{2\lambda} \right) \right)^{1/4}. \tag{11}$$

Next, to rescale the time, we define $\tau_{\eta,\lambda} : [0, \infty) \to [0, \infty)$ by

$$\tau_{\eta,\lambda}(t) = \int_{s=0}^{t} \frac{\eta^2}{R_{\eta,\lambda}(s)^4} \mathrm{d}s = \frac{1}{2} \ln \left( 1 + (e^{4\eta\lambda t} - 1) \frac{\eta}{2\lambda} \right) \tag{12}$$

where the second equality follows from a direct calculation. It is easy to show that $\tau_{\eta,\lambda}^{-1}(T) = \frac{\ln\left( \frac{2\lambda}{\eta}(e^{2T}-1)+1 \right)}{4\eta\lambda}$. Then we have the following theorem (see Appendix E for the proof), which says the rescaled version of SDE approximation Equation (3) admits the following limiting diffusion Equation (13), where $\{W(t)\}_{t\geq 0}$ is the standard Brownian motion in $\mathbb{R}^\Xi$.

$$\mathrm{d}Y(t) = -\frac{1}{2}Y(t)\mathrm{d}t + \frac{1}{2}\partial^2\Phi(Y(t))[\Sigma(Y(t))]\mathrm{d}t + \partial\Phi(Y(t))\sigma(Y(t))\mathrm{d}W(t) \tag{13}$$

**Theorem 4.1.** *Under Assumption 2.1 and 2.3, let $X_{\eta,\lambda}(0) \equiv x_{\mathrm{init}} \in U$ for all $\eta, \lambda > 0$ in SDE (3). Let $R_{\eta,\lambda}(t)$ and $\tau_{\eta,\lambda}(t)$ be defined in (11) and (12). If SDE (13) has a global solution $Y$ in $U$ with $Y(0) = \Phi(x_{\mathrm{init}})$, then $\frac{X_{\eta,\lambda}(\tau_{\eta,\lambda}^{-1}(T))}{R_{\eta,\lambda}(\tau_{\eta,\lambda}^{-1}(T))}$ converges in distribution to $Y(T)$ as $\eta\lambda \to 0$ with $\eta < 2\lambda < c$ for some constant $c$.*

**Remark 4.2.** *The additional constraint $\eta < 2\lambda$ when $\eta\lambda \to 0$ can be relaxed to $\eta < 2C\lambda$ for any constant $C > 0$. It suffices to note that for SDE (3), $X_{\eta,\lambda}$ with $X_{\eta,\lambda}(0) = x_{\mathrm{init}}$ and $X_{C^{-1/2}\eta,C^{1/2}\lambda}$ with $X_{C^{-1/2}\eta,C^{1/2}\lambda}(0) = C^{-1/4}x_{\mathrm{init}}$ have the same trajectories, up to a rescaling of $C^{-1/4}$. This is equivalent to replace $\eta/(2\lambda)$ by $\eta/(2C\lambda)$ in (11).*

## 4.2 Discrete-time Analysis for SGD+WD

Now, we proceed to analyze SGD+WD by mimicking the continuous-time behavior. Specifically, we view $R_{\eta,\lambda}(k)$ as an approximation of the norm of $x_{\eta,\lambda}(k)$, and consider the rescaled version of Equation (2) denoted by $\hat{x}_{\eta,\lambda}(k) = x_{\eta,\lambda}(k)/R_{\eta,\lambda}(k)$. Next, we introduce the time rescaling through the $\tau_{\eta,\lambda}(\cdot)$ defined in Equation (12) and denote $\tilde{t} = \tau_{\eta,\lambda}(t)$, so $t = \tau_{\eta,\lambda}^{-1}(\tilde{t})$. Now define $\widetilde{X}_{\eta,\lambda}(\tilde{t}) := \hat{x}_{\eta,\lambda}(\lfloor t \rfloor)$, and it can be shown that (see Appendix E for the derivation)

$$\widetilde{X}_{\eta,\lambda}(\tilde{t}) = \widetilde{X}_{\eta,\lambda}(\tilde{t}) + \int_{\tilde{s}=0}^{\tilde{t}} -\nabla L(\widetilde{X}_{\eta,\lambda}(\tilde{s}))\mathrm{d}A_{\eta,\lambda}(\tilde{s}) - \int_{\tilde{s}=0}^{\tilde{t}} \widetilde{X}_{\eta,\lambda}(\tilde{s})\mathrm{d}B_{\eta,\lambda}(\tilde{s})$$
$$- \int_{\tilde{s}=0}^{\tilde{t}} \sigma(\widetilde{X}_{\eta,\lambda}(\tilde{s}))\mathrm{d}Z_{\eta,\lambda}(\tilde{s}) \tag{14}$$

where $A_{\eta,\lambda}$, $B_{\eta,\lambda}$ and $Z_{\eta,\lambda}$ are defined by

$$A_{\eta,\lambda}(\tilde{t}) = \sum_{i=1}^{\lfloor \tau_{\eta,\lambda}^{-1}(\tilde{t}) \rfloor} \frac{\eta}{R_{\eta,\lambda}(i)R_{\eta,\lambda}(i+1)}, \tag{15}$$

$$B_{\eta,\lambda}(\tilde{t}) = \sum_{i=1}^{\lfloor \tau_{\eta,\lambda}^{-1}(\tilde{t}) \rfloor} \eta\lambda - (1 - \eta\lambda)\left( \frac{R_{\eta,\lambda}(i)}{R_{\eta,\lambda}(i+1)} - 1 \right), \tag{16}$$

$$Z_{\eta,\lambda}(\tilde{t}) = \sum_{i=1}^{\lfloor \tau_{\eta,\lambda}^{-1}(\tilde{t}) \rfloor} \frac{\eta\sqrt{\Xi}}{R_{\eta,\lambda}(i)R_{\eta,\lambda}(i+1)}\left( \mathbb{1}_{\xi_i} - \frac{1}{\Xi}\mathbb{1} \right). \tag{17}$$

The convergence of $A_{\eta,\lambda}$, $B_{\eta,\lambda}$ and $Z_{\eta,\lambda}$ are summarized in the following lemma.

**Lemma 4.3.** *Let $A_{\eta,\lambda}, B_{\eta,\lambda}$ and $Z_{\eta,\lambda}$ be as defined in Equation* (15), (16) *and* (17) *respectively. Then as $\eta\lambda \to 0$ with $\eta < 2\lambda < c$ for some constant $c$, it holds that (1) $A_{\eta,\lambda}$ increases infinitely fast, (2) $B_{\eta,\lambda}(t)$ converges to $\frac{t}{2}$ in uniform metric, and (3) $Z_{\eta,\lambda}$ converges in distribution to $(I_\Xi - \frac{1}{\Xi}\mathbb{1}\mathbb{1}^\top)^{1/2}W$ in uniform metric where $\{W(t)\}_{t \geq 0}$ is the $\Xi$-dimensional standard Brownian motion.*

Therefore, $\widetilde{X}_{\eta,\lambda}(\tilde{t})$ is also a valid Katzenberger process. Applying Theorem 2.10 yields:

**Theorem 4.4.** *Under Assumption 2.1 and 2.3, let $x_{\eta,\lambda}(0) \equiv x_{\mathrm{init}} \in U$ for all $\eta, \lambda > 0$ in SGD+WD* (2)*. Let $R_{\eta,\lambda}(t)$ and $\tau_{\eta,\lambda}(t)$ be defined in* (11) *and* (12)*. If SDE* (13) *has a global solution $Y$ in $U$ with $Y(0) = \Phi(x_{\mathrm{init}})$, then for any $T > 0$, $\frac{x_{\eta,\lambda}(\lfloor \tau_{\eta,\lambda}^{-1}(T) \rfloor)}{R_{\eta,\lambda}(\lfloor \tau_{\eta,\lambda}^{-1}(T) \rfloor)}$ converges in distribution to $Y(T)$ as $\eta\lambda \to 0$ with $\eta < 2\lambda < c$ some constant $c$.*

# 5 Mixing to Equilibrium

Now we proceed to study the ergodicity of the limiting diffusion (13). Omitted proofs of this section are delayed to Appendix F.

Due to the nature of the scale invariant of the loss $L$, we only care about the direction of $Y(t)$, i.e., $Y(t)/\|Y(t)\|_2$. To study the ergodicity of the normalized diffusion process, we need some additional assumptions on $\Gamma$ and the noise covariance. For any $r > 0$, define $\Gamma_r := \Gamma \cap \{x \in \mathbb{R}^D : \|x\|_2 = r\}$. We assume that $\Gamma_1$ is compact manifold to ensure the existence of stationary distribution of the limiting diffusion process. We also need to assume that $\Gamma_1$ are connected (so is $\Gamma$) for the uniqueness of the stationary distribution.

**Assumption 5.1.** $\Gamma_1$ is compact and connected.

We further assume that the noise is non-degenerate on the manifold of local minimizers, so that as a Markov chain the limiting diffusion is irreducible.

**Assumption 5.2** (Controllability). *For each $x \in \Gamma$, $\mathrm{span}(\{\partial\Phi(x)\sigma_i(x)\}_{i=1}^\Xi) = T_x(\Gamma_{\|x\|_2})$.*

**Assumption 5.3.** $\mathrm{tr}(\Sigma(\cdot))$ is an analytic function on $\mathbb{R}^D \setminus \{0\}$ and $\Gamma$ is an analytic manifold.

Now we are ready to state our main result in this section, which is Theorem 5.4. It is proved in two cases respectively in Appendices F.3 and F.4, depending on whether the trace of gradient covariance $\mathrm{tr}(\Sigma)$ is constant on $\Gamma_1$ or not. If it is, then the diffusion process essentially is on a $(D - M - 1)$-dimensional manifold (after suitable rescaling), $\Gamma_1$, just as in the analysis by Wang and Wang [25]. Otherwise, the situation becomes more complicated and the diffusion process is on $\Gamma$, a $(D - M)$-dimensional manifold, in which case we will need the analyticity assumption (Assumption 5.3).

**Theorem 5.4.** *Under Assumption 2.1, 2.3, 5.1, 5.2 and 5.3, starting from any initialization $Y(0) \in \Gamma$, the distribution of $\overline{Y}(t)$ converges to a unique stationary distribution $\pi$ on $\Gamma_1$ in total variation.*

Our main result on the fast mixing of SGD+WD and its SDE approximation follows from a direct combination of the convergence of the SGD+WD iterates proved in Theorem 4.4 and Theorem 5.4. Here note that as $\Gamma_1$ is compact, convergence in total variation implies convergence in distribution.

**Theorem 5.5** (Fast Mixing of SGD+WD). *Under Assumption 2.1, 2.3, 5.1, 5.2 and 5.3, let $x_{\eta,\lambda}(0) \equiv X_{\eta,\lambda}(0) \equiv x_{\mathrm{init}} \in U$ for all $\eta, \lambda > 0$ for SGD+WD* (2) *and SDE approximation* (3)*. For any $T > 0$, as $\eta\lambda \to 0$ with $\eta = O(\lambda)$ and $\lambda = O(1)$, both $\bar{x}_{\eta,\lambda}(\lfloor \frac{\ln(\frac{2\lambda}{\eta}(e^{2T}-1)+1)}{4\eta\lambda} \rfloor)$ and $\overline{X}_{\eta,\lambda}(\frac{\ln(\frac{2\lambda}{\eta}(e^{2T}-1)+1)}{4\eta\lambda})$ converge in distribution to the same distribution, denoted by $\mu_{T,x_{\mathrm{init}}}$. Moreover, for every $x_{\mathrm{init}} \in U$, $\mu_{T,x_{\mathrm{init}}}$ weakly converges to the same equilibrium distribution $\pi$ supported on $\Gamma_1$ as $T \to \infty$.*

# 6 Experiments

In this section, we first empirically verify our theory for the time scaling of the dynamics in a simple setting where our theory applies. We then show that the diffusion process exists during the training of PreResNet on CIFAR-10, and it has implicit bias towards better generalization.

## 6.1 Verification of Time Scaling

**Setting and Theoretical Prediction.** We train the following normalized linear model by $\ell_2$ regression: $F_z(x) = \langle \frac{x}{\|x\|_2}, z \rangle$, where $x \in \mathbb{R}^D$ is the model parameter, and $z \in \mathbb{R}^D$ is the input.

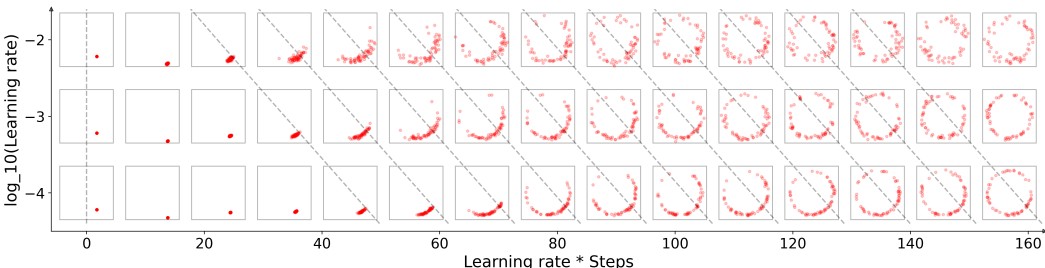

Figure 2: Scatter plots for projections of model parameters into the affine subspace containing the 1D zero-loss unit norm manifold. Models are trained with LR $\eta \in \{10^{-2}, 10^{-3}, 10^{-4}\}$ and WD factor $\lambda = 0.05$. Each small box in the figure contains 60 models that are trained with the same LR and the same number of steps. The $x$ axis indicates the product of the LR and the number of steps of each box; the $y$ axis indicates $\log_{10}(\text{LR})$ of each box. The dashed lines represent the time scaling $\frac{1}{2}\ln\left(1 + (e^{4\eta\lambda t} - 1)\frac{\eta}{2\lambda}\right) = T$ for different $T$'s (from 0 to 15.58), where $t$ is the number of steps in SGD+WD. The dynamics are consistent with time scaling suggested by our theory (Theorem 4.1).

Let $\{z_i\}_{i=1}^N$ be the input samples, and $y_i = F_{z_i}(x^*)$ be the target label for each $z_i$, for some $x^*$. The training loss is $L(x) = \frac{1}{2N}\sum_{i=1}^N (F_{z_i}(x) - F_{z_i}(x^*))^2$. We set $N = D - 2$, so the solution space $S = \{w : \langle w - \frac{x^*}{\|x^*\|_2}, z_i \rangle = 0, \forall i \in [N]\}$ is a 2-dimensional linear space. The manifold of unit-norm global minimizers $\Gamma_1$ is then equal to $S \cap \{x \mid \|x\|_2 = 1\}$. We generate $\{z_i\}_{i=1}^N$ randomly in a way that almost surely $x^*$ is not contained in the linear span of $\{z_i\}_{i=1}^N$. This implies that $M = \text{rank}(\nabla^2 L(x)) = D - 2$ on $\Gamma_1$ and that $\Gamma_1$ is a 1-dimensional manifold (a circle), thus Assumption 2.1 and 5.1 hold. During training, we set the loss at step $t$ as $L^{(t)}(x) = L(x) + \langle \frac{x}{\|x\|_2}, \epsilon_t \rangle$ where $\epsilon_t \overset{iid}{\sim} N(0, \hat{\sigma}^2 I_D)$ for some $\hat{\sigma}$. Then the SGD+WD update rule is

$$x_{\eta,\lambda}(k+1) = (1 - \eta\lambda)x_{\eta,\lambda}(k) - \eta\left(\nabla L(x_{\eta,\lambda}(k)) + \left(I_D - \frac{xx^\top}{\|x\|_2^2}\right)\cdot\frac{\epsilon_t}{\|x\|_2}\right). \qquad (18)$$

Let $\sigma(x) = \frac{\hat{\sigma}}{\|x\|_2}(I_D - \frac{xx^\top}{\|x\|_2^2})$, and the canonical SDE approximation is

$$dX_{\eta,\lambda}(t) = -\eta\nabla L(X_{\eta,\lambda}(t))dt - \eta\lambda X_{\eta,\lambda}(t)dt + \eta\sigma(X_{\eta,\lambda}(t))dW(t). \qquad (19)$$

For SDE approximation, we set $\Xi = D$, and thus Theorem 4.1 applies to Equation (19), suggesting that *the correct time scale (number of steps for SGD+WD) for the limiting dynamics is* $\tau_{\eta,\lambda}^{-1}(T) = \frac{\ln((2\lambda/\eta)(e^{2T}-1)+1)}{4\eta\lambda}$ *for each* $T \geq 0$. Furthermore, Assumption 5.3 holds because every term in this example is analytic. Assumption 5.2 holds because for any vector $v \in T_x(\Gamma_{\|x\|_2})$, we have $\langle x, v \rangle = 0$, and hence $\partial\Phi(x)\sigma(x)v = \partial\Phi(x)\frac{\hat{\sigma}}{\|x\|_2}v = \frac{\hat{\sigma}}{\|x\|_2}v$. Therefore, our main theorem (Theorem 5.5) predicts that *the limiting dynamics mix in* $\frac{\ln(\lambda/\eta)+O(1)}{4\eta\lambda}$ *steps*.

**Remark 6.1.** *For ease of demonstration, our main theorem (Theorem 5.5) is proved for SGD+WD with finitely many samples and thus does not directly apply to Equation (18). However, our analysis can be extended to the case of Gaussian noise in an straightforward way and the claim in Theorem 5.5 indeed holds for Equation (18).*

**Experimental Results.** In our experiments, we choose $D = 10$, $\overline{\sigma} = 0.3$, the WD factor $\lambda = 0.05$, and LR $\eta \in \{10^{-2}, 10^{-3}, 10^{-4}\}$. In Figure 2, we plot the projections of $\frac{x}{\|x\|_2}$ on the solution space for 60 different runs with identical initialization for each $\eta$. For each run, the only differences are the LR $\eta$ and/or the noise $\xi_t$. The dashed lines in the figure indicates our time scaling, i.e., $\frac{1}{2}\ln\left(1 + (e^{4\eta\lambda t} - 1)\frac{\eta}{2\lambda}\right) = T$. Figure 2 shows that time scaling of $O((\ln 2\lambda/\eta + T)/(\lambda\eta))$ fits the dynamics better, compared to $O(T/(\eta\lambda))$.

## 6.2 Limiting Diffusion on CIFAR-10

Beyond the toy example, we further study the limiting diffusion of PreResNet on CIFAR-10 [26]. We train a 32-layer PreResNet [27] with initial LR $\eta = 0.8$ and WD factor $\lambda = 5 \cdot 10^{-4}$. Unlike the normalized linear model, it is hard to visualize the model projection of PreResNet on the manifold. Instead, we choose the test accuracy of $\Phi(x_t)$ as a test function. In particular, we decay the LR to $10^{-3}$ to approximate gradient flow at different time $t$, and record the test accuracy after training 1000 more epochs. The results are shown in Figure 1 and Figure 3.

We observe that without LR decay, the train accuracy, test accuracy and parameter norm converge quickly after training 100 epochs; the test accuracy of $\Phi(x_t)$ converges much slower. It suggests that there exists a mixing process after reaching the manifold. Moreover, we observe the test accuracy of $\Phi(x_t)$ after convergence is significantly higher than $\Phi(x_t)$ at 100th epoch. It indicates that the mixing process is beneficial for generalization.

Our time scaling (12) suggests that the optimal step of LR decay grows no faster than $\tilde{\Omega}(1/\eta\lambda)$ as $\eta \to 0$. Unfortunately, (12) alone is not sufficient for deciding the optimal step for decaying LR as the mixing time $T$ for the continuous dynamics is unknown. A potential usage of time scaling (12) is to first estimate $T$ via another SGD run with a larger LR, which we leave for future work.

## 7    Related Work

**Normalization and Scale Invariance.** Previous works have analyzed the benefits of normalization layers from different viewpoints [28, 13, 29–50]. As noted before, normalization layers induce the scale invariance. It has been shown that scale invariance enables robust and efficient training of SGD+WD [51]. Scale invariance also brings about the interesting equivalence between the effect of WD and LR schedules [52, 19, 17]. Moreover, for SGD+WD with LR $\eta$ and WD factor $\lambda$, the parameter norm will converge to $(\lambda/\eta)^{1/4}$ [1, 53, 54], and this induces the intrinsic LR which is equal to $\eta\lambda$ [1]. These observations are crucial to our derivations in the current paper.

**Fast Equilibrium Conjecture.** Recently, Wang and Wang [25] proposed a spherical SDE model to approximate SGD+WD with constant LR. Using a novel adaption of Simon's theory, they justified the Fast Equilibrium Conjecture by showing that SGD+WD dynamics consists of three stages: descent ($O(1/\sqrt{\lambda\eta})$ time), diffusion ($O(1/(\lambda\eta)$ time) and tunneling ($O(e^{C/(\lambda\eta)})$ time). However, their analysis relies on the strong assumption of the minimizers of $L$ being isolated, which is against the empirical evidence that the level sets of deep learning loss functions are connected [55]. As a result, the diffusion phase shall bring no generalization benefit and cannot explain the improvement of final generalization if staying at training loss plateau for a longer time (see Figure 1). We allow the local minimizers to form a connected manifold, which can be viewed a generalization of the Morse function assumption [25], as an isolated minimizer is just a manifold of dimension 0.

Another common weakness of existing analyses in [1, 25] is that they only work for the SDE approximation (3), and do not apply to the actual discrete-time dynamics (2). In contrast, our results can handle both the continuous and discrete time dynamics under more reasonable assumptions.

**SDE Approximation.** Continuous-time tools such as SDE have been a popular lens for studying optimization algorithms including SGD [56–61]. Many interesting properties of SGD have been discovered through this approach [1, 62–65].

**Slow Dynamics of SGD Around Zero Loss Manifold.** Recent works [66, 16, 2] show that under the assumption that the minimizers locally connect as a manifold, SGD with label noise with small learning rate will move around the manifold after convergence, towards the direction of smaller trace of Hessian, at a very slow rate of $O(\eta^2)$ per step. Arora et al. [23] show that such slow dynamics on manifold can happen without stochastic gradient noise, if the update rule is non-smooth around the manifold of minimizers and GD enters Edge of Stability regime ([67]. Concretely, they show that normalized GD implicitly penalizes the largest eigenvalue of the Hessian at the rate of $O(\eta^2)$. Additional related works are deferred to Appendix A.

## 8    Conclusion and Future Work

We provide an SDE-based characterization for the limiting dynamics of SGD+WD for a scale invariant loss as $\eta\lambda \to 0$ with $\eta = O(\lambda)$ and $\lambda = O(1)$. Under some technical assumptions, we further show that the limiting diffusion converges to a unique stationary distribution. It leaves as future work to relax the technical assumptions. Another interesting and important direction for future work is to understand and characterize the benefit on generalization induced by the limiting diffusion.

## Acknowledgements

This work is funded by NSF, ONR, Simons Foundation, DARPA and SRC. ZL is also supported by Microsoft Research PhD Fellowship.

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
