# OpenReview forum: "Fast Mixing of Stochastic Gradient Descent with Normalization and Weight Decay"
_NeurIPS.cc/2022/Conference — NeurIPS 2022 Accept_

### Official Review · Reviewer_o9Wv · 2022-07-10

**Rating:** 8
**Confidence:** 4
**Soundness:** 4 excellent
**Presentation:** 4 excellent
**Contribution:** 4 excellent

**Summary:**

This papers theoretically proves the fast equilibrium conjecture that SGD for scale invariant loss mixes in  O(1/\lambda eta) steps by con sidering a new time rescaling limiting SDE. The limiting SDE of SGD without WD mixes much slower than SGD with WD justifies the benefit of weight decay for generalization performance.  The paper also provide some empirical results showing that the fast mixing obtains in asymptotic analysis does appear in pratical setting.


**Questions:**

As stated in weakness.

**Limitations:**

The major two limitations of the current results is:
1. The technical assumptions may not be satisfied.
2. The asymptotic approximation does not necessarily fully characterize SGD.

However, as I have stated in Strengths and Weakness, the current results are already enough for a good NeurIPS paper.

**Strengths And Weaknesses:**

Strengths:
Overall, this is an excellent paper that provides important theoretical results in understanding the behavior of SGD in training deep neural networks. More specifically,
Originality: The fast equilibrium conjecture was proposed to explains the interesting generalization performance as mentioned in line 33. As the name suggested, this conjecture has not been proved before. Therefore, the proof in this paper is original. Moreover, the new time rescaling SDE has never been studied in the existing literature, which is totally new to my best knowledge.
Quality: All the results in this paper is clearly stated and justified. I briefly checked the proof and it looks sound to me.
Clarity: A well written paper with main results highlighted.
Significance: The proof of this conjecture  provide us a reliable tool to understand SGD, which is theoretically significant.

Weakness:
1. It would be better to provide some intuitive explanations of the assumption made to prove the main result. Can these assumptions be satisfied in the realized case?
2. Beyond the proof of the conjecture, it would be more interesting to see what this mixing property can lead to some new conclusions. (Though I agree the content is already enough for one paper. )
3. Section 5.2 provide some discretization analysis. More work should be done on this, e.g., the discretization error.
4. Please correct typos in the paper, e.g., line 27 affect -> effect.

---

> ### Author Response · Authors · 2022-08-02
> **Response**
>
> We would like to thank the reviewer for the careful reading, thoughtful feedback, and strong support! Please see our responses to the questions and comments below.
>
> **Question**: Can the assumptions be satisfied in the realized case?
>
> **Answer**: The assumption 6.1, i.e., covariance on unit sphere has constant trace, is empirically justified in [Li et al, 2020]. The assumption 6.5, i.e., $\partial \Phi(x)\sigma(x)$ has maximal rank can be replaced by the lie algebra of $\{\partial \Phi(x)\sigma(x)\}_{i=1}^{\Xi}$ has maximal rank, which is much weaker. We state the former in the main paper for simplicity, but in the proof of appendix, we only use the latter. We will elaborate more on the validity and limitation of the assumptions in the future version.
>
>
> **Question**: Section 5.2 provide some discretization analysis. More work should be done on this, e.g., the discretization error.
>
> **Answer**: We think there is some misunderstanding. We did not approximate SGD using SDE nor prove for SDE only. The results in Section 5.2 hold directly for SGD without any heuristic approximation.
>
> **Question**: Please correct typos in the paper, e.g., line 27 affect -> effect.
>
> **Answer**: Thanks for pointing out the typo! We will carefully check these.
>
> *Reference*:
>
> - Li, Zhiyuan, Kaifeng Lyu, and Sanjeev Arora. "Reconciling modern deep learning with traditional optimization analyses: The intrinsic learning rate." Advances in Neural Information Processing Systems 33 (2020): 14544-14555.

---

> > ### Comment · Reviewer_o9Wv · 2022-08-08
> > **Thanks for the responses.**
> >
> > I do not have any other concern. Thanks! Though I do value the theoretical part of the paper, as suggested by other reviewers, it would be of great benefit to improve the quality of experiments.

---

### Official Review · Reviewer_Wb74 · 2022-07-10

**Rating:** 5
**Confidence:** 2
**Soundness:** 3 good
**Presentation:** 3 good
**Contribution:** 3 good

**Summary:**

This paper provided the dynamic convergence of SGD combined with the popular weight-decay technique (SGD-WD) for scaled-invariant loss functions (e.g., neural networks with normalization). Specifically, 1) the authors showed the iterates of SGD converge in distribution to some random variable in $\mathcal{O}\left(\frac{1}{\eta \lambda}\ln \left( \frac{2\lambda}{\eta}(e^{2T}-1) +1\right)\right)$ steps, where $\eta$ is step-size, $\lambda$ is the weight-decay parameter, and $T$ is the budget of the iterations. 2) In the regime of $\eta = \mathcal{O}(1/\lambda)$ when $(\eta, \lambda \rightarrow 0)$, the authors showed that SGD without WD has the same limiting diffusion as SGD with WD but its convergence is slower. 3) Under the assumptions on the manifold and the noise covariance, the authors partially proved the faster equilibrium conjecture of [26].

**Questions:**

The questions are addressed above.

**Limitations:**

Yes, they have mentioned their limitations.

**Strengths And Weaknesses:**

In general, the paper is clear and well-written. As I know, the interplay of normalization, gradient noise, and weight decay of SGD focused in this paper is a very important and interesting problem. The study of their interplay is beyond the optimization theory. The main contribution is that the authors provided the limiting dynamics convergence of SGD with weight decay and gave empirical evidence to verify their results.  However, I am not an expert in the dynamic theory of SDE. I am afraid that I can not give a fair assessment of the theoretical parts. But I checked the numerical results carefully and gave my concerns.

**About the experiments on the toy example:**
- In line 262, Equality (24) has not been mentioned in the main content. The authors may use (10) instead.  The same problem appears in line 290.
- in line 267, the authors said that as long as no $z_i$ is in the same direction with $x^{\ast}$, $\cdots$, then Assumption 6.4 holds. My question is how to ensure that no $z_i$ is in the same direction with $x^{\ast}$. In general, it is difficult to gain knowledge about $x^{\ast}$.
**The results on CIFAR10:**
- From Figure 1(a), the values of test accuracy and training accuracy change a lot and the algorithm is unstable. I am afraid it is hard to say that the algorithm has converged around 100 epochs. I guess because the initial step-size $\eta = 0.8$ is too large. I am curious how do you select the initial step size? Are you based on the best performance of the algorithm?
- As I know, a lot of papers studied that decaying the step size improves the performance. I think the interesting result is that the authors claimed that they can use the time scaling in (24) as an upper bound to decide the proper time of LR decay. Have you used this time scaling (24) to decide when to drop the step size in your experiments? I did not see that the selection of 100 epochs is based on their theory.
- Recently, there are some studies about step-decay step size, which is very efficient in practice. This step size keeps a constant at first and then drops by a constant factor after some iterations. For example, in [1], the authors analyzed this step-decay step size with the knowledge of the total number of iterations $T$. And they drop the step size every the same time period. I think the interesting thing is that the authors can use this theory of time scaling (24) to drop the step size adaptively. The authors have claimed this point but I am not very sure if the formula of (24) can be used in practice and achieve good performance in deep neural networks. I think more experiments are needed if the authors verify this claim.

[1] Wang, Xiaoyu, Sindri Magnússon, and Mikael Johansson. "On the convergence of step decay step-size for stochastic optimization." Advances in Neural Information Processing Systems 34 (2021): 14226-14238.

---

> ### Author Response · Authors · 2022-08-02
> **Response**
>
> We would like to thank the reviewer for the careful reading and thoughtful feedback.
>
> We would like first to clarify the focus of this study is not about **efficient optimization**, but the **generalization** of the training algorithm, which is often contradictory to the optimization efficiency in our settings (SGD on normalized networks). For example, Figures 1 and 3 show that staying at the plateau of large learning rate longer can significantly improve the generalization after decay, but also makes optimization less efficient. We did not perform experiments using the time scaling to decide the LR decay in this paper.
>
> Please also see our general response above, and our responses to other questions and comments below.
>
> **Question**: The initial step-size $\eta=0.8$ is too large. How do you select the initial step size?
>
> **Answer**: Our learning rate $0.8$ is indeed comparable to the standard $0.1$ learning rate for SGD with $0.9$ momentum. We enlarge the LR by $8$ times to compensate for turning off the momentum and achieve similar training behavior (including final test accuracy).
>
> **Question**: The selection of 100 epochs in Figure 1 does not seem to be based on theory.
>
> **Answer**: We believe there is a misunderstanding in our experiments. We do not suggest LR decay at the 100th epoch in Figure 1. Instead, our point is that even after the train accuracy stops increasing and plateaus (which roughly happens after 100 epochs), the iterate may still make meaningful movements which could increase the test accuracy after LR decay.
>
> **Question**: In line 262, Equality (24) has not been mentioned in the main content. The authors may use (10) instead. The same problem appears in line 290.
>
> **Answer**: Thanks for pointing this out! We have modified the equation reference in the revision.
>
> **Question**: How to ensure that no $z_i$ is in the same direction with $x^∗$? In general, it is difficult to gain knowledge about $x^∗$.
>
> **Answer**: If $z_i$ are randomly generated, as we have done in our experiments, the probability that $x^*$ is contained in the linear span of $\{z_i\}_{i=1}^{\Xi}$ is $0$. (Here we made a typo in the paper: “As long as no $z_i$ is in the same direction with $x^*$” should be “As long as $x^*$ is not contained in the linear space of $z_i$”. We have fixed this in the revision.)

---

> > ### Comment · Reviewer_Wb74 · 2022-08-07
> > **Response to authors**
> >
> > Thank the authors for their response.
> >
> > However, I can not agree with your response related to Figure 1.  In Figure 1(a), the training accuracy oscillates between 0.8 and 0.93, and test accuracy changes between 0.75 and 0.85. The baseline algorithm is very unstable. I am afraid that your statements based on Figure 1 are not reliable. I agree that the test accuracy may improve when the learning rate decays. I believe that some recent papers have already shown this.
> >
> > From a practical viewpoint, I think the most interesting thing about this paper is the time-scaling (10), which can be used to guide when to drop the step size. But the authors said that they did not use this time-scaling (10) to decide the LR decay in their numerical experiments. From this point, the paper is less interesting for me.

---

> > > ### Author Response · Authors · 2022-08-08
> > > **Clarifications**
> > >
> > > We thank Reviewer Wb74 for the feedback. To clarify the concerns on our CIFAR-10 experiments, we have revised Figures 1(a) and 3(a) in our paper and added Figure 4 in the appendix.
> > >
> > > - The training accuracy in the original Figure 1(a) and 3(a) was plotted for each training step and has been changed to accuracy per epoch in the revision. As observed from the new plot, the training curve across epochs is smooth, indicating the training of the algorithm is indeed stable. The oscillation of the test accuracy before the first LR decay is common in practice (see figures in this popular GitHub repo https://github.com/bearpaw/pytorch-classification/blob/master/utils/images/cifar.png). We also compare our large learning rate setting with the typical setting with $0.1$ learning rate and $0.9$ momentum in Figure 4.
> > >
> > > - Since the main contribution of this paper is theoretical, i.e., proving the Fast Equilibrium Conjecture, we do not claim any practical improvement implied by our theory. Indeed, equation (10) alone is not enough to decide the optimal step for LR decay, because there is an unknown parameter $T$. Rather, equation (10) describes the relationship between optimal steps for the decay of different LR. In other words, one needs another run of different LR to infer the parameter $T$. We deeply agree with the reviewer that deciding the optimal step for LR decay using our theory is an important question and we leave it for future work.

---

> > > > ### Comment · Reviewer_Wb74 · 2022-08-09
> > > > **Response to Authors**
> > > >
> > > > Thank the authors for their changes.  After the discussion with the authors, I also did the numerical experiments on CIFAR10 related to Figure 1.  The performance is evaluated over every epoch.
> > > >
> > > > I would like to mention that in the original Figure 1(a), the x-axis is the number of Epoch. It is natural to think that the training and test accuracy are evaluated per epoch. But the authors said that their original training accuracy of Figures 1(a) and 3(a) is evaluated in every training step. They changed the plot across epochs, then it is smooth. I am confused about  1) Do the authors plot the test accuracy in every training step or epoch in the same figure?  2) If the authors used epoch as the x-axis, why do they evaluate the training accuracy in every training step in the previous plots?   3) As in my experiments, the training accuracy still has big oscillation.
> > > >
> > > > In lines 290-292 of Section 7.2, I noticed that the authors claim that ``Although it is unclear whether the test accuracy of $\Phi(x_t)$ converges at the same rate as the diffusion process, one can always use the time scaling in (10) as an upper bound to decide the proper time of LR decay." But in the numerical experiments, the authors did not use (10) to guide LR decay.
> > > >
> > > > As I said before, I can not make a fair assessment of the theoretical part. From the numerical part and the responses of the authors, I am still not convinced.

---

> > > > > ### Author Response · Authors · 2022-08-09
> > > > > **Follow-up discussion with Reviewer Wb74**
> > > > >
> > > > > We thank the reviewer for the feedback! Let us provide further clarifications as follows:
> > > > >
> > > > > - We evaluate the model on the test dataset after every epoch of the training, so the test accuracy is plotted **per epoch** in the current Figures 1(a) and 3(a). We apologize for the confusion caused by the previous figures, where we plot the training accuracy of each step because we use the default logging of PyTorch Lightning (that training accuracy is only logged per step). We thank the reviewer again for bringing this up so we are able to correct this.
> > > > >
> > > > > - For the reproducibility of the experiment, we will release the code before camera-ready if the paper is accepted. For quick verification, we recommend running the following command on [this GitHub repo](https://github.com/bearpaw/pytorch-classification) that allows training PreResNet-32 on CIFAR-10 with 0.8 LR and no momentum: `python cifar.py -a preresnet --depth 32 --epochs 1000 --schedule 1000 --lr 0.8 --momentum 0 --wd 5e-4`. One should observe a smooth train accuracy curve along with an oscillating test accuracy curve similar to our Figure 1(a).
> > > > >
> > > > > - Again, we would like to clarify that we never claim that we use equation (10) to guide LR decay in our experiments. As mentioned in our last response, it is possible to use equation (10) to decide the time of LR decay if $T$ is known. However, it is not straightforward to infer the value of $T$, and we leave it for future work. We will definitely clarify this point in the paper.

---

> > > > > > ### Author Response · Authors · 2022-08-09
> > > > > > **Revision and details about the quick verification**
> > > > > >
> > > > > > We deleted the original statement in line 290-292 and added one paragraph at the end of Sec. 7.2 to further clarify about using equation (10) to decide the LR decay time.
> > > > > >
> > > > > > For your information, here is a list of changes might take to run [this GitHub repo](https://github.com/bearpaw/pytorch-classification) with recent Pytorch versions:
> > > > > > - Remove `async=True` in line 248 of `cifar.py`.
> > > > > > - Replace `loss.data[0]` with `loss.item()`, `prec1[0]` with `prec1.item()` and `prec5[0]` with `prec5.item()` in line 257-259, 314-317 of `cifar.py`.
> > > > > > - Replace `view` with `reshape` in line 16 of `utils/eval.py`.

---

### Official Review · Reviewer_QxFR · 2022-07-16

**Rating:** 6
**Confidence:** 1
**Soundness:** 3 good
**Presentation:** 3 good
**Contribution:** 3 good

**Summary:**

The main contributions of the paper are 1) providing a characterizing SDE limiting the SGD+WD by using a time-rescaling technique so that the results by Katzenberger are applicable 2) showing that SGD without WD for stochastic scale invariant loss has the same limiting dynamics as SGD+WD, but is exponentially slowed down 3) under the assumption of all minimizers forming a compact manifold and noise being non-degenerate in the tangent space of the manifold, the authors show that from any initialization, the limiting diffusion process converges to a unique stationary distribution 4) lastly, the authors conduct experiments to empirically show that these results hold.

**Questions:**

1) What are the implications of this paper for the training of ML/DL models?


**Limitations:**

Mentioned in weaknesses.

**Strengths And Weaknesses:**

Strengths:
The authors are able to show the limiting dynamics for SGD+WD and devise a novel time rescaling scheme to make the results of [1] applicable as they claim that the results of [2] does not hold since there is no true minimizer in the usual sense.

Weaknesses:
It seems that the time rescaling approach is the only notable contribution of the paper. The derivation of the SDE and other results seem to follow from [2]. While there seem to be novelties, the framework to analyze these dynamics have been provided by [2] so on that ground the contribution seems to be marginal.

[1] G. S. Katzenberger. Solutions of a stochastic differential equation forced onto a manifold by a large drift. The Annals of Probability, pages 1587–1628, 1991.
[2] Z. Li, T. Wang, and S. Arora. What happens after sgd reaches zero loss?–a mathematical framework. arXiv preprint arXiv:2110.06914, 2021.

---

> ### Author Response · Authors · 2022-08-02
> **Response**
>
> We would like to thank the reviewer for the careful reading and thoughtful feedback. Please see our general response above, and our responses to other questions and comments below.
>
> **Question**: It seems that the time rescaling approach is the only notable contribution, which is still marginal.
>
> **Answer**:  We respectfully disagree.
>
> First, normalization layers are crucial for achieving best performance in practice, but the limiting dynamics of SGD on normalized networks have never been studied before. The previous results by [Li et al., 21] do not apply to this case because scale invariant loss with weight decay does not admit any minimizers and thus does not satisfy their assumptions. We are the first to point out the correct time scaling for studying limiting dynamics and provide a rigorous proof as well as experimental verification.
>
> Second, the time rescaling approach is definitely non-trivial, especially in the general case where the result of [Li et al, 21] does not apply. As mentioned by Reviewer o9Wv, “the new time rescaling SDE has never been studied in the existing literature, which is totally new to my best knowledge”.
>
> Third, besides the new scaling rule, we also analyze the mixing process of limiting dynamics and prove the Fast Equilibrium Conjecture under certain assumptions.
>
> *Reference*:
>
> - Li, Zhiyuan, Tianhao Wang, and Sanjeev Arora. "What Happens after SGD Reaches Zero Loss?--A Mathematical Framework." In International Conference on Learning Representations, 2021.

---

> > ### Comment · Reviewer_QxFR · 2022-08-06
> > **Response to the authors**
> >
> > Dear authors,
> >
> > Thank you for your thoughtful response. However, I still have a lingering question about the time scaling technique. The technique already exists for differential inclusion methods, such as 3.3 in [3]. While the time rescaling technique used in this submission is different, I feel that the technique is not completely new. Can the authors give some justifications for this? I would be much appreciated it.
> >
> >
> > [3] Davis, D., Drusvyatskiy, D., Kakade, S. et al. Stochastic Subgradient Method Converges on Tame Functions. Found Comput Math 20, 119–154 (2020).

---

> > > ### Author Response · Authors · 2022-08-08
> > > **Follow-up discussion with Reviewer QxFR**
> > >
> > > We first thank Reviewer QxFR for bringing this nice paper to our attention. **We would like to clarify that our time rescaling technique is fundamentally different from the time-shifting method in [Davis et. al, 2020]**. More specifically:
> > > 1. They first define the continuous linear interpolation of the discrete iterates to be x(t), and then consider the **time-shifted dynamics** given by $x^{\tau}(t) = x(\tau + t)$ as $\tau \to \infty$. In contrast, we consider the rescaled iterates $\bar x_{\eta, \lambda}(\lfloor\frac{\ln(\frac{2\lambda}{\eta}(e^{2T}-1)+1)}{4\eta\lambda}\rfloor)$ (note here the trajectory is only _right continuous_ in $T$), where there is no time shifting, but rather we are trying to view the dynamics of the iterates with the **time ‘accelerated’**, which explicitly depends on the step size $\eta$ and weight decay factor $\lambda$.
> > > 2. In [Davis et. al, 2020], the purpose of considering the time-shifted $x^\tau$ is to study the convergence property of the stochastic subgradient methods, which is completely different from our focus. Our goal is to study the movement of SGD+WD when the iterates are close to the manifold of local minimizers (i.e., the implicit bias), so there is no issue of convergence since the initialization is assumed to be close to the local minimizers.
> > > 3. Moreover, [Davis et. al, 2020] consider the time-shifted $x^\tau$ in the limit of $\tau \to \infty$, where $\tau$ does NOT depend on the problem parameters like the step size. On the contrary, **our choice of time-rescaling is aware of and motivated by the unique properties of the SGD+WD iterates**. As discussed in line 194~200, the time rescaling is inspired by the scaling of the norm of the iterates. To the best of our knowledge, this has never been studied in the existing literature.
> > >
> > > _Reference_:
> > > - Davis, D., Drusvyatskiy, D., Kakade, S. et al. Stochastic Subgradient Method Converges on Tame Functions. Found Comput Math 20, 119–154 (2020).

---

> > > > ### Comment · Reviewer_QxFR · 2022-08-09
> > > > **Response to the authors**
> > > >
> > > > Thank you for your detailed explanation. I will increase the score for this paper.

---

### Official Review · Reviewer_fu5c · 2022-07-24

**Rating:** 6
**Confidence:** 2
**Soundness:** 3 good
**Presentation:** 3 good
**Contribution:** 3 good

**Summary:**

This paper studies the mixing time of scale invariant models whose parameters are governed by a stochastic differential equation that approximates SGD + weight decay. Under certain regularity assumptions, the paper shows convergence of parameters to a unique distribution with a rate
 $$ \Theta( \frac{T+ln( \lambda/\eta) }{ \eta \lambda}) $$

**Questions:**


If the trace of the noise covariance matrix is upper bounded by a constant, could we still have similar results?

Although the paper critically rely on scale invariance, would we expect any similar theorems if the model had homogeneity degree L > 0?


**Limitations:**

Yes

**Strengths And Weaknesses:**

Under the setting of scale invariant models whose parameters are governed by a SDE approximation of SGD + WD, the paper proves the direction of SGD iterates converge in distribution to a solution of an SDE. The main technical strategy is to employ 1) time rescaling 2) scaling factor that closely mimics the normalization factor for the parameters.

As in  Z. Li (2021), time rescaling is used to ensure that the main workhorse (Katzenberger theorem) can be applied. Essentially, by rescaling time by $$ O(t/\eta^2) $$, the negative gradient loss becomes the dominant term in the SDE (for small learning rate eta) which results in the parameters being close to manifold \Gamma.

Furthermore, the paper shows a cool corollary that shows that scale-invariant models with no weight decay converge exponentially slower.


Weaknesses:
The main weaknesses are the strong technical assumptions (e.g. constant noise covariance trace, asymptotic results, smoothness assumptions) but understandable given the difficulty of the problem.

---

> ### Author Response · Authors · 2022-08-02
> **Response**
>
> We would like to thank the reviewer for the careful reading and thoughtful feedback. Please see our responses to the questions and comments below.
>
> **Question**: If the trace of the noise covariance is upper bounded by a constant, could we still have similar results?
>
> **Answer**: The first part of our main result – the limiting dynamics under time scaling $\tilde{\Theta}(1/\eta\lambda)$ still holds. However, the second part about mixing certainly needs a lower bound for noise, otherwise the mixing cannot happen. Instead, as shown by the recent work [Lyu et al., 2022], when there is no gradient noise, GD on scale invariant loss with weight decay will track some deterministic flow on the manifold which decreases the `spherical sharpness’ of Hessian.
>
> **Question**: Would we expect any similar theorems if the model had homogeneity degree L > 0 instead of scale invariance?
>
> **Answer**: The usage of scale invariance and weight decay makes the training objective admit no minimizers, and thus previous results [Li et al., 2021] cannot apply. If the network is L-homogeneous, then similar results in terms of limiting dynamics are immediate by applying [Li et al., 2021], though the time scaling is different, i.e., $\Theta(1/\eta^2)$ steps, while ours is $\tilde{\Theta}(1/\eta\lambda)$.
>
> *Reference*:
>
> - Lyu, Kaifeng, Zhiyuan Li, and Sanjeev Arora. "Understanding the Generalization Benefit of Normalization Layers: Sharpness Reduction." arXiv preprint arXiv:2206.07085 (2022).
>
> - Li, Zhiyuan, Tianhao Wang, and Sanjeev Arora. "What Happens after SGD Reaches Zero Loss?--A Mathematical Framework." In International Conference on Learning Representations, 2021.

---

### Author Response · Authors · 2022-08-02
**General Response**

We thank reviewers for their time and thoughtful reviews. We have responded to questions from each reviewer and revised the paper accordingly. Here, we would like to highlight the following question asked by both Reviewer QxFR and Reviewer Wb74.

**Question**: What is the implication of our theory for deep learning practice?

**Answer**: It's often observed that staying at the plateau of train accuracy for a longer period improves test accuracy of SGD after LR decay in practice (see our Figure 1). In such a case, our theory suggests the number of steps before LR decay should be $\tilde{\Theta}(1/\eta\lambda)$ for the best generalization after decay, which better explains experimental observation in [Li et al., 2020] than existing theory [Shi et al, 2020], which suggests that the mixing time (and thus steps needed before LR decay) is $e^{-\Theta(1/\eta)}$.

For example, as an application of our theory, if we shrink either LR or weight decay factor by 100 times, we should increase the training budget by roughly 100 times for similar generalization performance. See Figure 2 in [Li et al., 2020] for a better illustration.

*Reference*:

- Li, Zhiyuan, Kaifeng Lyu, and Sanjeev Arora. "Reconciling modern deep learning with traditional optimization analyses: The intrinsic learning rate." Advances in Neural Information Processing Systems 33 (2020): 14544-14555.

- Shi, Bin, Weijie J. Su, and Michael I. Jordan. "On learning rates and schr\" odinger operators." arXiv preprint arXiv:2004.06977 (2020).

---

### Meta-Review · Area_Chair_vQB2 · 2022-08-21

**Recommendation:** Accept
**Confidence:** Certain

**Metareview:**

This paper continues a line of works studying SGD via a similar SDE, this time employing a variety of tools not used before, for instance time-rescaling and normalization layers.  The reviewers on one hand were concerned at times that this was incremental, but also uncovered a variety of interesting contributions.  As such, I feel this paper is a clear accept, and am excited to see it in the conference.  That said, the discussions between reviewers and authors were quite detailed and I feel the authors could greatly strengthen their presentation by carefully adjusting for them, making their intended message more clear for future readers.

**Award:**

No

---

### Decision · Program_Chairs · 2022-09-14

Accept